# Association of Dietary Nitrate, Nitrite, and N-Nitroso Compounds Intake and Gastrointestinal Cancers: A Systematic Review and Meta-Analysis

**DOI:** 10.3390/toxics11020190

**Published:** 2023-02-17

**Authors:** Monireh Sadat Seyyedsalehi, Elham Mohebbi, Fatemeh Tourang, Bahareh Sasanfar, Paolo Boffetta, Kazem Zendehdel

**Affiliations:** 1Department of Medical and Surgical Sciences, University of Bologna, 40138 Bologna, Italy; 2Cancer Research Center, Cancer Institute, Tehran University of Medical Sciences, Tehran 1419733141, Iran; 3Department of Oncology, Lombardi Comprehensive Cancer Center, Georgetown University, Washington, DC 20007, USA; 4Stony Brook Cancer Center, Stony Brook University, Stony Brook, NY 11794, USA; 5Cancer Biology Research Center, Cancer Institute, Tehran University of Medical Sciences, Tehran 1419733141, Iran

**Keywords:** neoplasms, organ, nitrate, nitrite, esophageal neoplasms, gastric neoplasms, intestinal neoplasms, pancreatic neoplasms, organic chemicals

## Abstract

N-nitroso compounds (NOCs) are a class of chemical carcinogens found in various environmental sources such as food, drinking water, cigarette smoke, the work environment, and the indoor air population. We conducted a systematic review and meta-analysis to investigate the links between nitrate, nitrite, and NOCs in food and water and the risk of gastrointestinal (GI) cancers, including esophageal cancer (EC), gastric cancer (GC), colorectal cancer (CRC), and pancreatic cancer (PC). A systematic search of the literature in Scopus, PubMed, Google Scholar, Web of Science, ScienceDirect, and Embase was performed for studies on the association between NOCs in drinking water and food sources and GI cancers. Forest plots of relative risk (RR) were constructed for all the cancer sites and the intake sources. The random-effects model was used to assess the heterogeneity between studies. Forty articles were included after removing duplicate and irrelevant articles. The meta-analysis indicated that the intake of high dose vs. low dose of these compounds was significantly associated with the overall GI cancer risk and nitrite (RR = 1.18, 95% CI = 1.07–1.29), and N-nitrosodimethylamine (NDMA) (RR = 1.32, 95% CI = 1.06–1.65). We found that dietary nitrite intake increased GC (RR = 1.33, 95% CI = 1.02–1.73), and EC (RR = 1.38, 95% CI = 1.01–1.89). Additionally, dietary NDMA intake increased the risk of CRC (RR = 1.36, 95% CI = 1.18–1.58). This meta-analysis provides some evidence that the intake of dietary and water nitrate, nitrite, and NOCs may be associated with GI cancers. In particular, dietary nitrite is linked to GC and EC risks and dietary NDMA intake is associated with CRC.

## 1. Introduction

In 2020, cancer accounted for nearly 10 million deaths and nearly one in six deaths worldwide. Gastrointestinal (GI) cancers, with an estimation of approximately 5 million new cases and 3.5 million deaths worldwide, accounted for half of the cancer burden in 2020 [1,2]. The age-adjusted incidence rate (ASR) and mortality rate (ASMR) were equal to 19.5 and 9.7 per 100,000 for colorectal cancer (CRC) in both genders, respectively. The corresponding rates were 11.1 and 7.7 per 100,000 for gastric cancers (GC), 6.3 and 5.6 for esophageal cancer (EC), and 4.9 and 4.5 for pancreatic cancer (PC), respectively [1].

In addition to non-modifiable risk factors such as the demographic characteristics, family history, and genetic predisposition, some preventable risk factors are well established for GI cancer, including tobacco smoking, alcohol consumption, *H pylori* infection, high body mass index, low physical activity, and dietary factors [3,4,5,6].

The N-nitroso compounds (NOCs) are a broad class of chemical carcinogens that exist in various environmental sources such as food, drinking water, cigarette smoke, the work environment, and the indoor air population, although intrinsic sources are also available for crucial needs of our bodies [7,8,9,10,11,12]. The NOCs act as alkylating agents and may react with DNA to cause mutations leading to carcinogenicity [13,14,15]. N-nitrosodimethylamine (NDMA) is one of the NOCs found in human food, predominantly in processed/cured meats and smoked/salted fish [13]. Additionally, ingested nitrate is reduced to nitrite by the bacterial flora in the mouth and digestive tract. Subsequently, nitrite may react with amines, amides, and other nitrosation precursors in the gastrointestinal tract to form the NOCs under physiological conditions [11]. The main sources of nitrate are vegetables (beetroot, spinach, and cabbages, etc.) and the main sources of nitrite are animal foods (processed and smoked food) [8,16]. Drinking water is another source of nitrate in most countries, resulting from the overuse of chemicals or the improper disposal of human and animal waste, including fertilizers, feedlots, industrial and food processing waste [9].

Several experimental and epidemiological studies have shown that nitrate, nitrite and NDMA can affect human health [10], including an association cancer [17,18,19,20,21,22]. However, the results on the associations between GI cancers and these compounds are inconsistent. We performed a systematic review and meta-analysis to study the associations between the intake of these three compounds from food and water and the risk of GI cancers, including esophageal, gastric, colorectal, and pancreatic cancers.

## 2. Materials and Methods

### 2.1. Data Sources, Search Strategy, and Selection Criteria

The online database searches were performed in January 2022. Searches were undertaken for English-language peer-reviewed publications on the association of nitrate, nitrite, and NOCs and the risk of GI cancers between 1990 to the present. The databases included Scopus, PubMed, Google Scholar, Web of Science, ScienceDirect, and Embase. The search strategy was designed using MeSH terms like “Gastrointestinal Cancers”, “Digestive System Cancers”, “Organic Chemicals”, “Nitroso Compounds”, “Nitrate”, “Nitrite”,”NDMA”, and “Epidemiologic Studies”. Based on our searches on the databases, a total of 12,750 articles were retrieved, including 4539 articles for GC, 2204 for EC, 6673 for CRC, and 2243 for PC. A total of 40 studies were included after the title, abstract, full-text evaluations, and quality assessment [23,24,25,26,27,28,29,30,31,32,33,34,35,36,37,38,39,40,41,42,43,44,45,46,47,48,49,50,51,52,53,54,55,56,57,58,59,60,61,62] were undertaken. Figure 1 shows the PRISMA flow diagram of the literature searches and the study selection process. The inclusion criteria were as follows; case-control and cohort studies reporting either a relative risk (RR) or an odds ratio (OR) for the associations between GI cancers and the consumption of nitrate, nitrite, and NOCs from drinking water or food sources.

### 2.2. Data Extraction and Quality Assessment

The study screening and quality assessment were conducted by three researchers (MSS, FT, and BS), and in the event of any discrepancies, a referee (EM) intervened. The data extraction file contained the demographic characteristics of the article such as the author’s name, the year of publication, the title, the type of study, the country, as well as the design characteristics, including the sample size, the sampling method, the source of the population, the type of controls, the type of cancer, the source of nitrate, nitrite, and the NOCs (water/food/vegetable/fruit/all), and the method of data collection (e.g., food frequency questionnaire). Finally, the effect size measures, including the relative risks (RRs) for the cohort studies and the odds ratios (ORs) for the case-control studies and their 95% CI were abstracted.

### 2.3. Quality Assessment

Quality assessment of the included articles (case-control or cohort studies) was accomplished using the Newcastle-Ottawa Scale (NOS) [63]. For case-control studies, the checklist contained the definition and selection of cases and controls, comparability, ascertainment of exposure, and the method of ascertainment for the cases and the controls. The checklist of cohort studies included the situation of the exposed and non-exposed cohort, comparability, assessment of outcome, and the duration and adequacy of follow-up of the cohort study. Based on the NOS, the selection score could result in a maximum of four stars, the comparability scores a maximum of two stars, and the outcome/exposure score a maximum of three stars [64].

### 2.4. Statistical Analysis

The relationships between nitrite, nitrate, or NMDA intake and the risk of GI cancers were examined based on the effect size measurements and the corresponding 95% CIs of each study. Because of the rare disease assumption, the ORs are assumed to approximate the RRs [65]. Heterogeneity (Het.) among studies was evaluated by the Q test, based on the variation across studies rather than within studies, and the I^2^ statistic (the percentage of variance in a meta-analysis that is attributable to study heterogeneity) [66]. The pooled effect of nitrate, nitrite, and NOCs, in the case of significant heterogeneity (PQ test > 0.10 and I^2^ > 60%), was estimated using a random-effects model [67]. All analyses were completed using the STATA version 14.0 (Stata, College Station, TX, USA). Given that approximately 82% of the studies reported combined results for both genders, we focused our analysis on these results. Due to the high variation of nitrogen types, we conducted separate analyses on nitrate, nitrite, and NMDA intake among the studies. In order to estimate the impact of the dose-response relationship and to describe the magnitude of the response, we calculated the high/moderate doses of the nitrogen types and compared it to the low dose as a reference group in each study. A stratified analysis was conducted by gender (male and female), food sources (plant and animal), study design (case-control and cohort studies), and sub-sites of CRC (colon and rectum). Since most studies have been conducted in the United States (48%), stratification based on the country was ineffective. The *p*-value of heterogeneity was assessed using a sub meta-analysis. Moreover, publication bias was examined by creating a funnel plot and a regression asymmetry test [68,69].

## 3. Results

A total of 40 independent studies were included in the meta-analysis, including 27 case-control studies [23,24,25,26,27,28,31,33,34,35,36,37,38,39,41,42,43,45,51,52,54,55,56,57,60,61] and 13 cohort studies [29,30,32,40,46,47,48,49,50,53,58,59,62]. Details on these studies were provided in Appendix A. These studies reported a total of 13 risk estimates for CRC [34,36,42,46,49,51,53,54,55,56,59,62], 22 for GC [23,24,25,27,28,29,30,31,32,33,34,35,38,39,40,41,44,45,47,49,50,57], seven for EC [26,35,44,47,49,50,60], and five for PC [37,48,52,58,61]. The studies included in the meta-analysis had a minimum score of 7 out of 9 stars in the NOS.

### Meta-Analysis

The meta-analysis was repeated for each cancer site based on the nitrate, nitrite, NDMA and the source intake.

The findings revealed a statistically significant relationship between the highest vs. the lowest level of nitrite intake from the food sources and EC risk (RR = 1.38, 95% CI = 1.01–1.89). There were no statistically significant excess risks from the food sources of nitrate, NDMA, or the water sources of nitrate intake. We observed significant heterogeneity for studies of the food sources of NDMA (I^2^ = 80.2%, P-heterogeneity = 0.025), the food intake of nitrate (I^2^ = 89.4%, P-heterogeneity = 0.000), and the food intake of nitrite (I^2^ = 73.8%, P-heterogeneity = 0.004) (Figure 2). There were no significant results according to gender and food source (plant or animal) among the nitrate and nitrite intake groups (Table 1 and Table 2).

We found a significant association between the highest vs. the lowest level of nitrite consumption and GC (RR = 1.33, 95% CI = 1.02–1.73) (I^2^ = 92.2%, P-heterogeneity = 0.000). Additionally, there were significant heterogeneities for studies of the food intake of NDMA (I2 = 94.4%, P-heterogeneity = 0.000), the food intake of nitrate (I^2^ = 93.2%, P-heterogeneity = 0.000), and the water intake of nitrate (I^2^ = 93.5%, P-heterogeneity = 0.000) (Figure 3). We found non-significant differences in the association of GC and nitrite and nitrate intakes stratified for other factors (Table 1 and Table 2).

The association between the highest vs. the lowest levels of nitrate, nitrite, and NDMA and CRC risk only showed that the intake of NDMA from the food increased the risk of CRC significantly (RR = 1.36, 95% CI = 1.18–1.58). No statistically significant heterogeneity across studies was observed for CRC in relation to NDMA intake, but heterogeneity was statistically significant for studies of the food nitrate intake (I^2^ = 73.4%, P-heterogeneity = 0.002), the food nitrite intake (I^2^ = 77.2%, P-heterogeneity = 0.001), and the water intake of nitrate (I^2^ = 98%, P-heterogeneity = 0.000) (Figure 4). Stratified analyses according to the study type, the gender, the topography and the food sources (plant, animal) revealed no significant differences due to these variables (Table 1 and Table 2).

We found no significant associations between PC in the highest vs. the lowest levels of nitrate, nitrite, and NDMA. However, we observed significant heterogeneity for the food intake of nitrate (I^2^ = 80.7%, P-heterogeneity = 0.000), and nitrite (I^2^ = 93.5%, P-heterogeneity = 0.000) (Figure 5). No significant differences were found by study types, genders, topographies, and food sources (plants, animals) (Table 1 and Table 2).

## 4. Discussion

We found limited high-quality research on nitrate, nitrite, NOCs, and cancer risk, most likely due to the difficulty of quantifying these compounds from different sources worldwide. Despite this limitation, the literature suggests that dietary and water sources may be risk factors for GI cancers, in particular, nitrate intake from water sources and diet for CRC, nitrite intake from diet for GC, and NDMA intake for CRC and EC [14,15,56,59,70].

Nitrate and nitrite as precursors of the NOCs (e.g., NDMA) are suspected of playing a key role in cancer carcinogenesis through the induction of DNA-damaging metabolites, like aldehydes and alkyldiazonium ions, which could cause cancerous lesions in cells [14,15,16,58,70]. Several in vivo and epidemiological studies reported that different factors might contribute to the carcinogenic effect of nitrate, nitrite, and NOCs, including the dose and the type of the compounds, the cancer sub-site, the morphology of the tumor, the type of diet source (animal, plant, or water), the gender, the exposure time, the cooking methods, the effect of seasonal rainfall on water composition and the different fertilizer compounds used in different areas [62]. For example, consuming fresh fruits and vegetables high in vitamins, essential minerals, and antioxidants such as vitamins C and E, as well as reducing meat, fatty foods, and processed foods were found to improve health. This may be important in modifying any harmful effects of dietary nitrates and nitrites on particularly susceptible tissues in the digestive system [58,71].

There was heterogeneity in the study designs and the methodologies, which may explain the differences in the results of the published studies. The high heterogeneity observed in our meta-analyses may be due to the variety of confounders used in the models as well as the differences between the study settings, the population characteristics, the compound ranges, and the estimation methods. Except for age and gender, other confounders were less consistently controlled for, particularly dietary factors and specific risk factors, such as *H pylori* infection in the GC studies [72].

### 4.1. Esophageal Cancer

The number of reviews for EC was limited. A meta-analysis study by Lie et al. [19] in 2016 found no significant association between dietary nitrite and nitrate and EC, and a study conducted by Essien et al. [18] in 2020 reported similar results based on water nitrate. We found that dietary nitrite and animal sources of nitrate were positively associated with a higher risk of EC, but NDMA was not associated with EC. These results may indicate that the source and the amount of the intake are important determinants of cancer risk. Animal sources of food (mainly processed meats) contain amines and amides that are necessary precursors for endogenous nitrosation [19]. Although other sources like vegetables may contain these compounds, they are also rich in antioxidants, which may decrease their harmful effect and explain their protective role in cancer risk [71].

### 4.2. Gastric Cancer

In 2016, Lie and others found an inverse association between dietary nitrate consumption and GC, and a borderline association between dietary nitrite consumption and GC [19]. In 2015, Peng Song et al. [22] found that nitrates from the diet can reduce GC risk, but nitrites and NDMA increased the risk. Fei-Xiong and colleagues reported that dietary nitrate decreased, and nitrite intake increased GC risk [21]. A meta-analysis carried out in 2020 by Essien et al. found no significant association between water-source nitrate intake and GC risk. [18]. In the present study, we found that dietary nitrite intake increased the risk of GC, but the associations were not significant for all kinds of dietary and water sources of nitrate and NDMA intake. Unexpectedly, we reported a significant inverse association between dietary nitrate and GC in males. In addition to the source of compounds, other GC risk factors such as *H.pylori* infection may increase the NOCs product level and increase the risk of GC [39].

### 4.3. Colorectal Cancer

Previous studies on association between the NOCs and CRC are inconsistent. In 2016, one systematic review reported that dietary nitrite/nitrate intake was not significantly linked with CRC [19], however, another review in 2020 reported that dietary nitrate consumption was linked to a higher risk of CRC [18]. However, the intake of dietary nitrite and nitrate from the drinking water was not associated with CRC risk. Another review in 2020 revealed that nitrate from water sources was significantly associated with CRC [20]. Our study showed that dietary NDMA intake increased the risk of CRC, but nitrite and nitrate sources were not significantly associated with CRC. The possible reasons for the association between NDMA and CRC may relate to the main source of intake, including meats, particularly bacon, hot dogs, and sausage, which are naturally high in amines. The association may be explained by the synergy between NDMA and amine groups which are found in protein structures [54].

### 4.4. Pancreatic Cancer

None of the previous reviews reported a significant relationship between dietary or water intake of nitrate, nitrite, and NDMA and the risk of PC [18,19]. Our results of the relation between water and dietary sources of nitrate, nitrite, and NDMA showed no significant association with PC.

The geographical distribution of the studies plays an important role in achieving reliable results. A majority of the studies included in our review were conducted in Europe (30%) and North America (48%), and we found no reliable studies from low- and middle-income countries (LMICs). As a result, our findings cannot be generalized to LMICs, where the amounts of nitrate, nitrite, and NOCs may vary based on the dietary patterns, the cooking methods, the water sources, the effect of seasonal rainfall on water compounds, and the fertilizer compounds [73]. Also, most studies on the water sources were of ecological design and therefore were not included in our analysis. A small number of studies in relation to several exposure/outcome combinations, and high heterogeneity in most of our meta-analyses were additional limitations. Furthermore, possible residual confounding like consuming diets and lifestyle factors and inadequate adjustment in several studies might have hampered the results of the published studies. However, the present systematic review and meta-analysis have several important strengths. We included the most updated articles that reported the associations for GC, EC, CRC, and PC exclusively. We also studied different sources of nitrate, nitrite, and NOCs (i.e., animal, plant, and water sources) intake.

## 5. Conclusions

In conclusion, our meta-analysis shows modest evidence on the association between dietary and water nitrate, nitrite, and NOCs intake and certain types of GI cancer risk. The dietary intake of nitrite is associated with GS and EC and the dietary NDMA intake is associated with CRC. In future studies, the influence of different compound types on cancer should be explored by considering the source intake, and gender differences, particularly in less studied geographical regions.

## Figures and Tables

**Figure 1 toxics-11-00190-f001:**
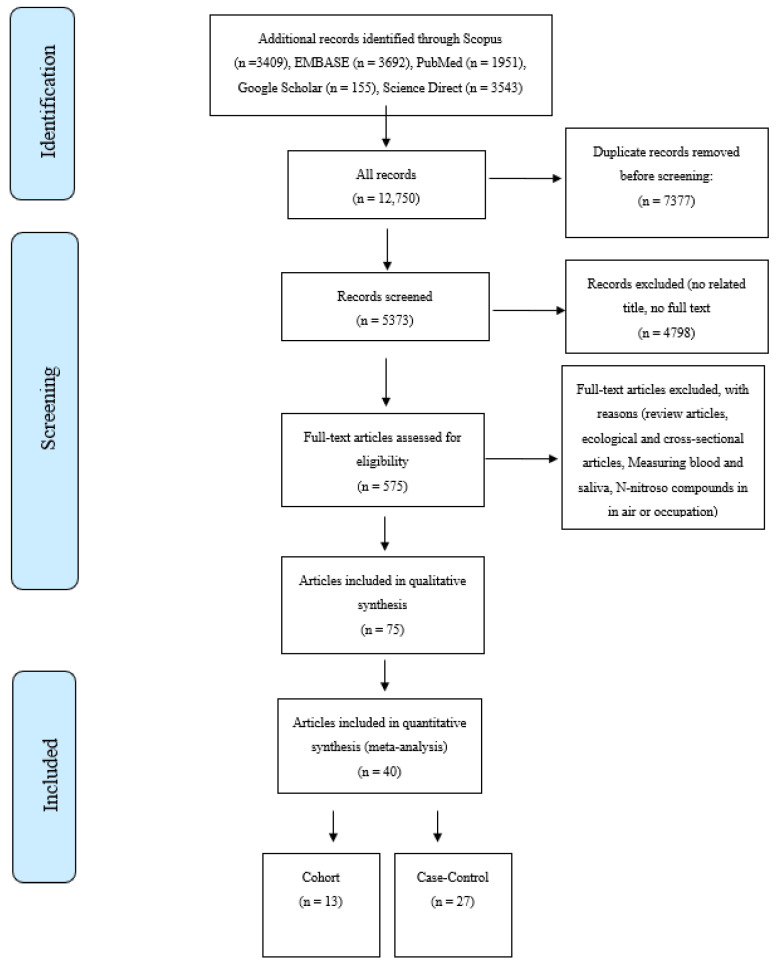
Selection process for studies included in the meta-analysis for the intake of nitrate, nitrite, and N-nitroso compounds from diet and water and the risk of gastrointestinal cancers.

**Figure 2 toxics-11-00190-f002:**
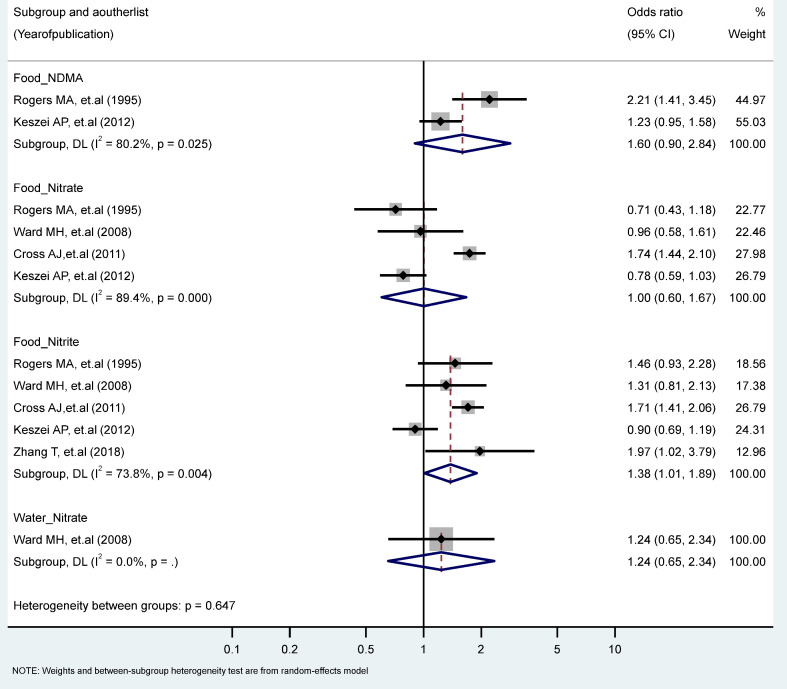
Forest plot (random-effects model) quantifying the relationships between NDMA, nitrite, and nitrate intake and esophagus cancer risk stratified by sources. List of references includes: [26,44,47,50,60].

**Figure 3 toxics-11-00190-f003:**
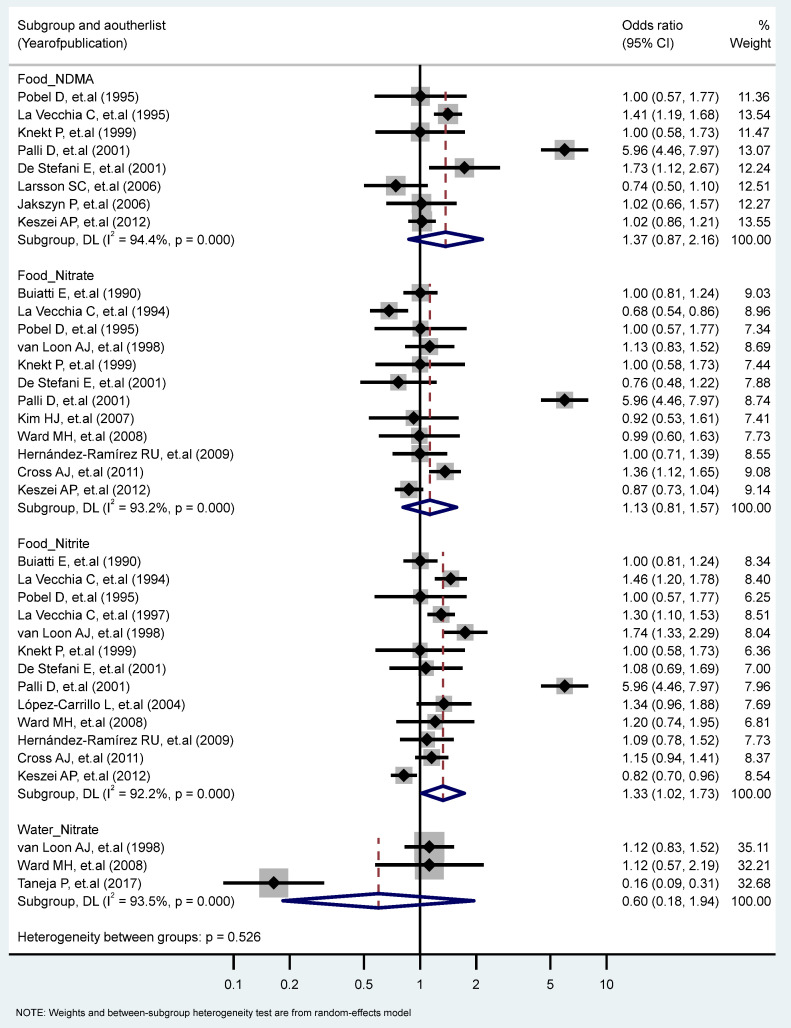
Forest plot (random-effects model) quantifying the relationships between NDMA, nitrite, and nitrate intake and gastric cancer risk stratified by sources. List of references includes: [23,24,25,27,30,32,33,34,38,39,40,41,44,45,47,50,57].

**Figure 4 toxics-11-00190-f004:**
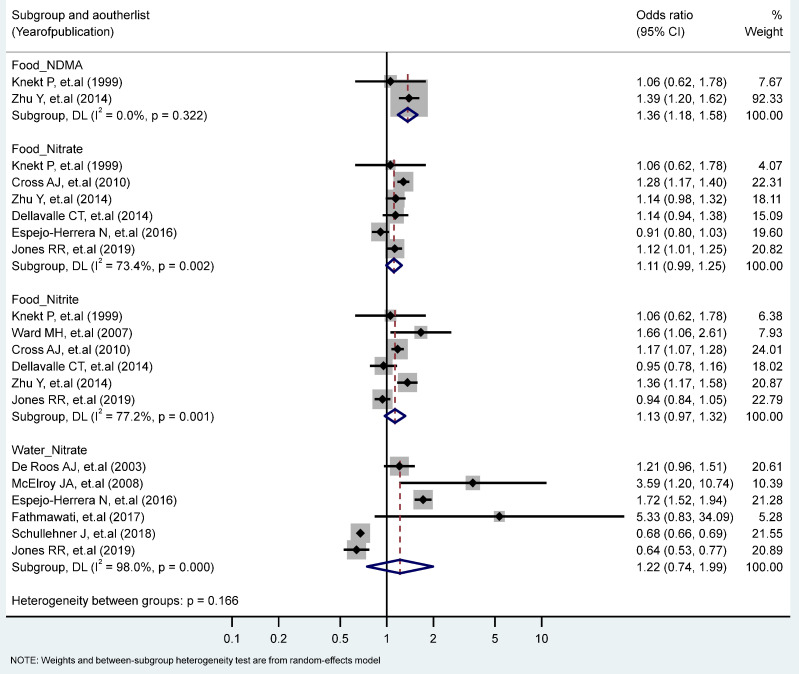
Forest plot (random-effects model) quantifying the relationships between NDMA, nitrite, and nitrate intake and colorectal cancer risk stratified by sources. List of references includes: [32,36,42,43,46,53,54,55,56,59,62].

**Figure 5 toxics-11-00190-f005:**
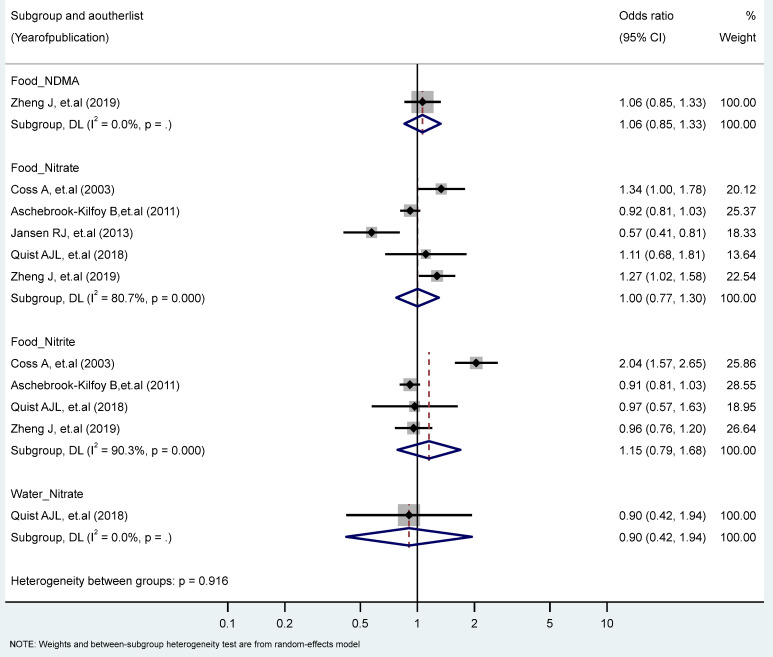
Forest plot (random-effects model) quantifying the relationships between NDMA, nitrite, and nitrate intake and pancreatic cancer risk stratified by sources. List of references includes: [37,48,52,58,61].

**Table 1 toxics-11-00190-t001:** Association between nitrate intake and selected gastrointestinal cancers stratified by gender, food sources, study type, and topography.

		Colorectal Cancer				Gastric Cancer				Esophageal Cancer				Pancreatic Cancer			
		N	RR (95% CI)	I^2^	P_b_	N	RR (95% CI)	I^2^	P_b_	N	RR (95% CI)	I^2^	P_b_	N	RR (95% CI)	I^2^	P_b_
gender	Male	0	-	-	-	1	0.78 (0.63–0.97)	-	-	1	0.84 (0.61–1.18)	-	-	2	0.86 (0.45–1.66)	9.82	0.002
	Female	0	-	-	-	1	1.08 (0.79–1.47)	-	-	1	0.62 (0.37–1.05)	-	-	2	1.50 (1.27–1.77)	0.01	0.917
P-heterogeneity		-				0.513				0.066				0.400			
Food Sources	Animal	3	1.12 (0.93–1.35)	11.70	0.003	3	1.06 (0.76–1.48)	6.28	0.043	1	1.73 (1.43–2.10)	-	-	2	0.83 (0.41–1.69)	12.12	0.000
	Plant	2	0.99 (0.76–1.29)	5.34	0.021	3	0.99 (0.77–1.28)	0.00	0.999	1	0.96 (0.57–1.60)	-	-	1	1.23 (0.99–1.53)	-	-
P-heterogeneity		0.368				0.884				0.294				0.206			
Study Type	Cohort	4	1.19 (1.10–1.29)	3.96	0.266	4	1.08 (0.83–1.39)	11.36	0.010	2	1.17 (0.53–2.57)	21.83	0.000	2	0.92 (0.82–1.04)	0.54	0.461
	Case_ control	2	1.01 (0.81–1.26)	5.12	0.024	8	1.14 (0.65–1.99)	148.92	0.000	2	0.82 (0.57–1.18)	0.67	0.412	3	1.00 (0.62–1.61)	17.46	0.000
P-heterogeneity		0.093				0.469				0.419				0.182			
Topography	Colon	5	1.07 (0.94–1.21)	14.93	0.005												
	Rectum	5	1.13 (0.96–1.33)	10.87	0.028												
p-heterogeneity		0.084															

N = number of studies; P_b_ = *p* value for heterogeneity.

**Table 2 toxics-11-00190-t002:** Association between nitrite intake and selected gastrointestinal cancers stratified by gender, food sources, study type, and topography.

		Colorectal Cancer				Gastric Cancer				Esophagus Cancer				Pancreatic Cancer			
		N	RR (95% CI)	I^2^	P_b_	N	RR (95% CI)	I^2^	P_b_	N	RR (95% CI)	I^2^	P_b_	N	RR (95% CI)	I^2^	P_b_
Food Sources	Animal	4	1.03 (0.89–1.19)	20.80	0.000	4	1.00 (0.81–1.23)	7.73	0.052	2	1.25 (0.67–2.33)	14.17	0.000	3	1.36 (0.88–2.10)	28.11	0.000
	Plant	2	0.95 (0.86–1.05)	0.01	0.937	3	1.10 (0.86–1.40)	0.24	0.888	1	1.31 (0.80–2.13)	-	-	2	0.85 (0.75–0.96)	0.86	0.353
P-heterogeneity		0.546				0.618				0.199				0.916			
Study Type	Cohort	4	1.02 (0.88–1.19)	10.40	0.015	3	1.30 (0.94–1.81)	6.62	0.036	1	1.70 (1.41–2.06)	-	-	2	0.91 (0.81–1.03)	0.04	0.835
	Case_ Control	2	1.38 (1.20–1.59)	0.68	0.409	9	1.42 (0.99–2.02)	111.4	0.000	3	1.49 (1.11–2.00)	0.97	0.616	2	1.39 (0.66–2.93)	18.75	0.000
P-heterogeneity		0.254				0.014				0.000				0.711			
Topography	Colon	4	1.08 (0.97–1.20)	6.47	0.091												
	Rectum	4	1.04 (0.85–1.29)	9.13	0.028												
P-heterogeneity		0.153															

N = number of studies; P_b_ = *p* value for heterogeneity.

## Data Availability

Not applicable.

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
