# Peer review of "Association of Dietary Nitrate, Nitrite, and N-Nitroso Compounds Intake and Gastrointestinal Cancers: A Systematic Review and Meta-Analysis"

_toxics, 2023, doi:10.3390/toxics11020190_

Round 1
Reviewer 1 Report
In their manuscript “N-nitroso compounds intake and gastrointestinal cancer risks: a systematic review ad meta-analysis”, Seyyedsalehi et al. try to evaluate the safety of ingested NOC from foodstuff and water for developing GI cancer.
While I appreciate the amount of work and analysis, I have difficulties to agree with parts of their work/ideas flow.
First, they need to explain more in details the relation between N-nitrosodimetyhylamine (NDMA) and nitrite/nitrate. Nitrite/nitrate are naturally occurring compounds in vegetables – such as celery, beets, green-leaf vegetables, and, interestingly, even fresh meat (without any additives, muscles in living humans contain rather large amount of nitrate – see Wylie et al. J Physiol 2019, 597:5565-5576). NDMA is not a natural foodstuff and there is a question about if, how and when would nitrite convert into NDMA.
Interestingly, consumption of high amounts of vegetables, such as in case of vegetarians/vegans does not lead to known pandemic of GI cancer (as an example see India where large portion of population is vegetarian). I therefore believe that the results of only weak pro-GI-cancer evidence due to nitrate/nitrite dietary consumption that authors found in their analysis should be more emphasized.
Could the studies included in their meta analysis be sub-divided into two separate groups that show evidence for NDMA and nitrate/nitrite?
Author Response
Response to the reviewer 1:
First, they need to explain more in details the relation between N-nitrosodimetyhylamine (NDMA) and nitrite/nitrate. Nitrite/nitrate are naturally occurring compounds in vegetables – such as celery, beets, green-leaf vegetables, and, interestingly, even fresh meat (without any additives, muscles in living humans contain rather large amount of nitrate – see Wylie et al. J Physiol 2019, 597:5565-5576). NDMA is not a natural foodstuff and there is a question about if, how and when would nitrite convert into NDMA.
Interestingly, consumption of high amounts of vegetables, such as in case of vegetarians/vegans does not lead to known pandemic of GI cancer (as an example see India where large portion of population is vegetarian).
I therefore believe that the results of only weak pro-GI-cancer evidence due to nitrate/nitrite dietary consumption that authors found in their analysis should be more emphasized.
Could the studies included in their meta analysis be sub-divided into two separate groups that show evidence for NDMA and nitrate/nitrite?
Dear reviewer: according to your comments and suggestions, the title of the paper has been changed to "Dietary Nitrate, Nitrite, and N-Nitroso Compounds Intake and Gastrointestinal Cancer Risk: a systematic review and meta-analysis", and the text has been revised regarding NOC definitions and their relationship to nitrate and nitrite. We added some explanations to the introduction and discussion.
We agree on the comment about the high intake of vegetables and their relation to GC. As we discuss in the paper, fruit and vegetables, including a group of vitamins and anti-oxidants, may affect NOC's carcinogenic effect and protect against GI cancer. Despite this, epidemiological studies from different geographic areas like India have limitations with regard to nitrate, nitrite, and NOCs.
One study: references number 53
Taneja P, Labhasetwar P, Nagarnaik P, Ensink JHJ. The risk of cancer as a result of elevated levels of nitrate in drinking water and vegetables in Central India. J Water Health, 2017, 15,602-614.
In response to the last comment, we reported by cancer type, specifically focusing on nitrates, nitrites, and NOCs, and clarified the source of intake.
Reviewer 2 Report
This is an interesting topic, given that France has just published a new report on nitrate/nitrite/NOCs and cancers. I appreciate the great effort to do the literature search and screening. However, I notice that there are some critical knowledge mistakes in the manuscript. In particular, nitrate and nitrite are not NOCs. This seems a big issue for this meta-analysis.
I did not read through it because of this issue. However, it would be great if the authors could correct those mistakes.

Author Response
Response to the reviewer 2:
Comments and Suggestions for Authors
This is an interesting topic, given that France has just published a new report on nitrate/nitrite/NOCs and cancers. I appreciate the great effort to do the literature search and screening. However, I notice that there are some critical knowledge mistakes in the manuscript. In particular, nitrate and nitrite are not NOCs. This seems a big issue for this meta-analysis.
I did not read through it because of this issue. However, it would be great if the authors could correct those mistakes.
Dear reviewer: according to your comments and suggestions, the title of the paper has been changed to "Dietary Nitrate, Nitrite, and N-Nitroso Compounds Intake and Gastrointestinal Cancer Risk: a systematic review and meta-analysis", and the text has been revised regarding NOC definitions and their relationship to nitrate and nitrite. We added some explanations to the introduction and discussion.
Reviewer 3 Report
The manuscript submitted by Seyyedsalehi et al provided a systematic review and meta-analysis of the association of N-nitroso compounds (NOCs) intake with gastrointestinal (GI) cancer risks. It appears that NOCs intake contributes to the incidence of some types of GI cancers, but to a relatively less extent, in the studies citied here. While this review can clearly provide useful information for GI cancer etiology studies, some revisions are needed to improve the quality of this manuscript.
1. In the Introduction, the authors claimed that “The three most common forms of NOCs are nitrite, nitrate, and NDMA. The main resources of these NOCs are …”However, there are some errors to be revised.
(1) N-nitroso compounds, by definition, should contain a nitroso group which thus exclude nitrate and nitrite from the class. Nitrate and nitrite salts in fact facilitate the formation of NOCs via facile nitrosation at the physiological conditions.
(2) There are many types of NOCs to which humans are commonly exposed (Gushgari, A.J.; Halden, R.U. Chemosphere 2018, 210, 1124–1136). The most prevalent NOCs are actually tobacco-derived nitrosamines such as NNN and NNK (Li Y; Hecht SS. Int. J. Mol. Sci. 2022, 23(9), 5109; https://doi.org/10.3390/ijms23095109). NDELA is the major NOC found in cosmetics and contributes much to the daily exposure to humans. In foods, NDMA occurred mostly at the highest concentrations, averaging at 2.2 ng/g; however, NDBA and NPYR also contributed greatly, averaging at 1.5 ng/g (Li Y; Hecht SS. Int. J. Mol. Sci. 2022, 23(9), 4559; https://doi.org/10.3390/ijms23094559).
(3) NDMA detected in H2O is considered to mainly result from chloramination disinfection (Mitch, W.A.; Sharp, J.O.; Trussell, R.R.; Valentine, R.L.; Alvarez-Cohen, L.; Sedlak, D.L. Environ. Eng. Sci. 2003, 20, 389–404; Beard, J.C.; Swager, T.M. J. Org. Chem. 2021, 86, 2037–2057.).
2. The early epidemiology studies with NOCs have been reviewed by the International Agency for Research on Cancer (IARC monograph volume Sup. 7, 1987 and volume 77, 2000). It would be good to mention these studies and then provide the updated data to this topic in the manuscript.
3. In the Discussion, I suggest to expand the content to other etiological factors of GI cancers. For example, it is now widely accepted that infectious bacteria (e.g., Helicobacter pylori) play an important role in gastric cancers (Yang L, et al. The Lancet Public Health, 2021, 6 (12), e888-896). The consumption of red meat has been linked to the incidence of colorectal cancers. For esophageal cancers, hot food and other eating habits along with the exposure of food-source NOCs are all potential causing factors to increase the incidence risk. It will be more informative to include the recent progress of the major etiological risk factors to these GI cancers and provide a clearer picture of the possible roles of NOCs in GI carcinogenesis to people who are not familiar to the relevant research areas.
Author Response
Response to the reviewer 3:
Comments and Suggestions for Authors
The manuscript submitted by Seyyedsalehi et al provided a systematic review and meta-analysis of the association of N-nitroso compounds (NOCs) intake with gastrointestinal (GI) cancer risks. It appears that NOCs intake contributes to the incidence of some types of GI cancers, but to a relatively less extent, in the studies citied here. While this review can clearly provide useful information for GI cancer etiology studies, some revisions are needed to improve the quality of this manuscript.
- In the Introduction, the authors claimed that “The three most common forms of NOCs are nitrite, nitrate, and NDMA. The main resources of these NOCs are …”However, there are some errors to be revised. (1) N-nitroso compounds, by definition, should contain a nitroso group which thus exclude nitrate and nitrite from the class. Nitrate and nitrite salts in fact facilitate the formation of NOCs via facile nitrosation at the physiological conditions. (2) There are many types of NOCs to which humans are commonly exposed (Gushgari, A.J.; Halden, R.U. Chemosphere 2018, 210, 1124–1136). The most prevalent NOCs are actually tobacco-derived nitrosamines such as NNN and NNK (Li Y; Hecht SS. Int. J. Mol. Sci. 2022, 23(9), 5109; https://doi.org/10.3390/ijms23095109). NDELA is the major NOC found in cosmetics and contributes much to the daily exposure to humans. In foods, NDMA occurred mostly at the highest concentrations, averaging at 2.2 ng/g; however, NDBA and NPYR also contributed greatly, averaging at 1.5 ng/g (Li Y; Hecht SS. Int. J. Mol. Sci. 2022, 23(9), 4559; https://doi.org/10.3390/ijms23094559). (3) NDMA detected in H2O is considered to mainly result from chloramination disinfection (Mitch, W.A.; Sharp, J.O.; Trussell, R.R.; Valentine, R.L.; Alvarez-Cohen, L.; Sedlak, D.L. Environ. Eng. Sci. 2003, 20, 389–404; Beard, J.C.; Swager, T.M. J. Org. Chem. 2021, 86, 2037–2057.).
Thanks for constructive comment. According to the comments and suggestions, the title of the paper has been changed to "Dietary Nitrates, Nitrites, and N-Nitroso Compounds Intake and Gastrointestinal Cancer Risk: a systematic review and meta-analysis", and the text has been revised regarding NOC definitions and their relationship to nitrate and nitrite. In the introduction and discussion sections, we added some explanations.
- The early epidemiology studies with NOCs have been reviewed by the International Agency for Research on Cancer (IARC monograph volume Sup. 7, 1987 and volume 77, 2000). It would be good to mention these studies and then provide the updated data to this topic in the manuscript.
Thanks. We used it in the introduction and discussion section.
- In the Discussion, I suggest to expand the content to other etiological factors of GI cancers. For example, it is now widely accepted that infectious bacteria (e.g., Helicobacter pylori) play an important role in gastric cancers (Yang L, et al. The Lancet Public Health, 2021, 6 (12), e888-896). The consumption of red meat has been linked to the incidence of colorectal cancers. For esophageal cancers, hot food and other eating habits along with the exposure of food-source NOCs are all potential causing factors to increase the incidence risk. It will be more informative to include the recent progress of the major etiological risk factors to these GI cancers and provide a clearer picture of the possible roles of NOCs in GI carcinogenesis to people who are not familiar to the relevant research areas.
We added some sentences to the discussion section regarding heterogeneity and possible role of cofactors.
Round 2
Reviewer 1 Report
My concerns had been addressed the best they could at the current state of the field. I have no additional comments.
Author Response
My concerns had been addressed the best they could at the current state of the field. I have no additional comments.
Answer: Thanks for your approval.
Reviewer 2 Report
It’s a huge effort to pull everything together. However, I don't think the authors' response to my previous comment is adequate enough. The title is all good now, but the literature search strategy, screening criteria and analyses should be amended if you include nitrate and nitrite.
Firstly, nitrates and nitrites should be included as keywords, IF you include nitrate and nitrite as your targets.
Secondly, it is quite straightforward to use MeSH terms in PubMed, but I may specify search terminologies for each database. "Cancer" could be more widely used than 'neoplasms' in Scopus, WoS and Embase. It is not very good to use the same terms in all databases (please ignore if you have used different terms. If so, please upload them as a supplementary file). On the other hand, there will be many more publications if you add nitrate and nitrite as keywords. There are systematic reviews on nitrate and nitrite already, but https://www.anses.fr/fr/content/r%C3%A9duire-l%E2%80%99exposition-aux-nitrites-et-aux-nitrates-dans-l%E2%80%99alimentation for your consideration.
Thirdly, some important NOC individuals should be included as keywords if your focus is mainly on NOCs, such as NDMA, NDEA (both are recognised as 2A, probably carcinogenic to humans), and Nitrosamines (ok for PubMed, but for other databases). I understand that there are only a few eligible RCTs/epidemiology studies on nitrosamines.
Other a few very minor suggestions:
Please update the PRISMA diagram (some numbers are missing), and caption.
There is a trace amount of NOCs in fruit and vegetables, and in most processed meat. Be careful when you add nitrate and nitrite in the main text.
P-heterogeneity or p-heterogeneity
Pg 11, One meta-analysis conducted in 2020 by Essien et al. showed...
Author Response
Response to Reviewer 2:
It’s a huge effort to pull everything together. However, I don't think the authors' response to my previous comment is adequate enough. The title is all good now, but the literature search strategy, screening criteria and analyses should be amended if you include nitrate and nitrite.
Firstly, nitrates and nitrites should be included as keywords, IF you include nitrate and nitrite as your targets. Answer: They were included as the keyword.
Secondly, it is quite straightforward to use MeSH terms in PubMed, but I may specify search terminologies for each database. "Cancer" could be more widely used than 'neoplasms' in Scopus, WoS and Embase. It is not very good to use the same terms in all databases (please ignore if you have used different terms. If so, please upload them as a supplementary file). On the other hand, there will be many more publications if you add nitrate and nitrite as keywords. There are systematic reviews on nitrate and nitrite already, but https://www.anses.fr/fr/content/r%C3%A9duire-l%E2%80%99exposition-aux-nitrites-et-aux-nitrates-dans-l%E2%80%99alimentation for your consideration. Thirdly, some important NOC individuals should be included as keywords if your focus is mainly on NOCs, such as NDMA, NDEA (both are recognised as 2A, probably carcinogenic to humans), and Nitrosamines (ok for PubMed, but for other databases). I understand that there are only a few eligible RCTs/epidemiology studies on nitrosamines.
Answer: We included nitrate, nitrite, and also NDMA in the strategy method, but we didn't add them to the text in the previous version and we used different terms in each database. You can review the full search strategy in appendix of this letter. In response to your comment, we added the terms “nitrate, nitrite, and MDMA” in the updated version. We also changed “neoplasm” to “cancer” in the text.
Other a few very minor suggestions:
Please update the PRISMA diagram (some numbers are missing), and caption. Answer: updated
There is a trace number of NOCs in fruit and vegetables, and in most processed meat. Be careful when you add nitrate and nitrite in the main text. Answer: Thanks for your attention. We checked the text regarding this issue.
P-heterogeneity or p-heterogeneity Answer: Updated.
Pg 11, One meta-analysis conducted in 2020 by Essien et al. showed... Answer: we updated the sentence.

Reviewer 3 Report
I think the authors did not address all the raised questions especially due to the incorrect definition of NOCs when they conducted the meta-analysis. I also would like to see a significant improvement of the discussion so that other epidemiological factors can be weighed to their appropriate roles in GI cancers. The authors have to make the conclusion fairly clear about the currently recognized leading causes to those cancers.
Author Response
Response to Reviewr3
I think the authors did not address all the raised questions especially due to the incorrect definition of NOCs when they conducted the meta-analysis.
Answer: We included nitrate, nitrite, and also NDMA in the strategy method, but we didn't add them to the text in the previous version and we used different terms in each database. You can review the full search strategy in appendix of this letter. In response to your comment, we added the terms “nitrate, nitrite, and MDMA” in the updated version. We also changed “neoplasm” to “cancer” in the text.
I also would like to see a significant improvement of the discussion so that other epidemiological factors can be weighed to their appropriate roles in GI cancers. The authors have to make the conclusion fairly clear about the currently recognized leading causes to those cancers.
Answer: We listed the main risk factors of GI cancer in the interdiction. We also discussed other risk factors for each cancer sites in the discussion. (Page: 11 and 12). And also, some sentences in the introduction section.

Round 3
Reviewer 2 Report
I appreciate the authors' work in amending the manuscript. It is much stronger now.
1. Figure 1 caption should be moved to the bottom of fig.
Reviewer 3 Report
I have no further comments to the revised version of the manuscript.